# Analysis and study of risk factors related to the progression of non-alcoholic fatty liver disease: A retrospective cohort study

JunRan Yang⬤, Zhenhua Zhou⬤*

Shuguang Hospital Affiliated to Shanghai University of Chinese Traditional Medicine, Shanghai, China

* jinghua1220@163.com

## Abstract

### Objective

Non-alcoholic fatty liver disease (NAFLD) is becoming increasingly prevalent worldwide. This study aimed to analyze the risk factors associated with NAFLD progression by collecting and evaluating clinical data of NAFLD patients, providing a scientific basis for its prevention and treatment.

### Methods

Clinical data of NAFLD patients from June 2015 to June 2016 were retrospectively collected, including gender, age, alanine aminotransferase (ALT), aspartate aminotransferase(AST), alkaline phosphatase(ALP),γ-glutamyltranspeptidase (GGT), triglycerides (TG), total cholesterol (TC), high-density lipoprotein (HDL), low-density lipoprotein (LDL), fasting blood glucose (FBG), and visceral fat area (VFA). All patients were stratified by gender and age, and logistic regression analysis was used to explore the risk factors for NAFLD disease progression.

### Results

ALT, TG, FBG, and VFA were identified as independent risk factors for NAFLD progression. Stratified analysis showed that in male patients, ALT, TG, and VFA were independent risk factors, whereas in female patients, TG, FBG, and VFA were identified as independent risk factors. Age-stratified analysis revealed that ALT, TG, and VFA were significant risk factors for progression in young and middle-aged patients. At the same time, age, ALT, TG, and FBG were substantial in elderly patients.

### Conclusion

Different risk factors should be closely monitored in sex- and age-specific populations to prevent NAFLD progression effectively.

**Data availability statement:** All relevant data are within the paper and its Supporting Information files.

**Funding:** Natural Science Foundation of Anhui Province (2023085MH293) provides clinical basic research ideas for fatty liver. Science and Technology Support Project of Science and Technology Innovation Action Plan of Shanghai Municipal Science and Technology Commission (NO.21S1900400) provides the preclinical research foundation of fatty liver. Anhui University Major Research Project (NO.2023AH040098) plays a role in data collection and analysis.

**Competing interests:** The authors have declared that no competing interests exist.

# 1. Introduction

Non-alcoholic fatty liver disease (NAFLD) is a metabolic stress-induced liver injury characterized by excessive fat accumulation in hepatocytes, occurring in the absence of excessive alcohol consumption or other overt hepatotoxic factors [1], with a global prevalence rate of up to 25% [2,3], which has replaced chronic viral hepatitis as the most common chronic liver disease in the world, also in China. Moreover, China has the highest NAFLD-related annual mortality rate in Asia [4]. The burden of NAFLD has increased dramatically over the past two decades as China's lifestyle has changed radically [5]. NAFLD has no obvious symptoms in the early stage, only weakness, dull pain, and other discomfort in the liver area, which is easy to ignore. It progresses to non-alcoholic steatohepatitis, liver fibrosis, cirrhosis, and even hepatocellular carcinoma if left untreated [6–8].

Currently, there are no approved first-line drugs for the treatment of NAFLD, and no established therapies for hepatoprotection or antifibrosis in NASH patients [9,10]. Lifestyle modifications and weight management remain the primary recommendations for preventing NAFLD, as outlined in current guidelines [11]. Given the progressive nature of NAFLD and the severe consequences of delayed prevention and treatment, early identification of factors contributing to disease progression is critical. While numerous studies have explored the risk factors for NAFLD onset, few have investigated the determinants of NAFLD progression using a hierarchy of controlled attenuation parameter (CAP) values. This study aims to analyze risk factors for NAFLD progression, focusing on liver function, blood lipid profiles, and body fat distribution. Furthermore, it seeks to identify independent risk factors contributing to disease progression, providing a scientific foundation for the early prevention and management of NAFLD.

# 2. Information and methods

## 2.1 General Information

The dates we accessed the data for research purposes were 11/06/2015. The study subjects were retrospectively selected from June 2015 to June 2016 in the Department of Hepatology, Shuguang Hospital, Shanghai University of Traditional Chinese Medicine. Inclusion criteria were patients diagnosed with fatty liver by ultrasound, patients with complete physical examination data, and patients aged ≥18. The diagnosis of fatty liver was based on the 2010 criteria of the Chinese Medical Association Hepatology Branch [12]. Fatty liver was diagnosed by the presence of two or more of the following abnormalities in abdominal ultrasonography: (1) increased echogenicity of the liver in the near field and diminished echogenicity in the far field; (2) liver parenchymal echogenicity greater than renal parenchymal echogenicity; (3) intrahepatic blood vessels and biliary structures were not displayed. The exclusion criteria were (1) patients with hepatitis, schistosomiasis, liver disease, or cirrhosis caused by viral hepatitis, drug toxicity, immune disorders, alcoholism, etc.; (2) patients suffering from severe primary diseases of the heart, kidney, lungs, endocrine, hematologic, metabolic, and gastrointestinal tracts; (3) subjects with poor compliance

to the study protocols or those who have recently participated in or are currently participating in, other clinical trials; (4) lactating women, pregnant women, or women preparing for pregnancy. We performed a history review of patients with fatty liver disease to confirm the diagnosis of NAFLD in patients who met the following criteria. (1) Alcohol consumption of no more than 20g per day (2) Exclusion of viral hepatitis, drug-induced liver disease, Wilson's disease, total parenteral nutrition, autoimmune liver disease, and other special conditions that may cause fatty liver. (3) The diagnosis of abdominal ultrasound meets the diagnostic criteria formulated by the Chinese Society of Hepatology. A total of 545 patients were diagnosed with NAFLD by ultrasound at the Department of Hepatology, Shuguang Hospital, Shanghai University of Traditional Chinese Medicine, from June 2015 to June 2016, of which 215 were excluded according to the above inclusion and exclusion criteria, resulting in a final sample of 330 cases. The study followed the principles expressed in the Declaration of the World Medical Association of Helsinki the International Ethical Guidelines for Biomedical Research Involving Human Subjects (GIOMS, Geneva, 1993), and Chinese clinical research regulations. The study plan was approved by the Medical Ethics Committee of Shuguang Hospital, Shanghai University of Traditional Chinese Medicine(2015-399-27-01). Informed consent was obtained from all subjects for all investigations, and informed consent forms were signed.

(Fig 1 illustrates the screening process).

Based on the controlled attenuation parameter (CAP), the liver was categorized into mild, moderate, and severe non-alcoholic fatty liver disease (NAFLD). CAP values were measured using the Echosens FibroScan-520 device, employing transient elastography.During the examination, the patient lay supine on the examination bed with their hands positioned above their head to expose the abdomen and maximize the right intercostal space. The right intercostal region, between the anterior axillary line and the mid-axillary line, was selected as the examination site. The probe was carefully placed perpendicular to the skin at the designated location.The procedure was conducted by an experienced intermediate-level physician with 30 years of expertise in this technique. For each patient, at least 10 measurements were taken from

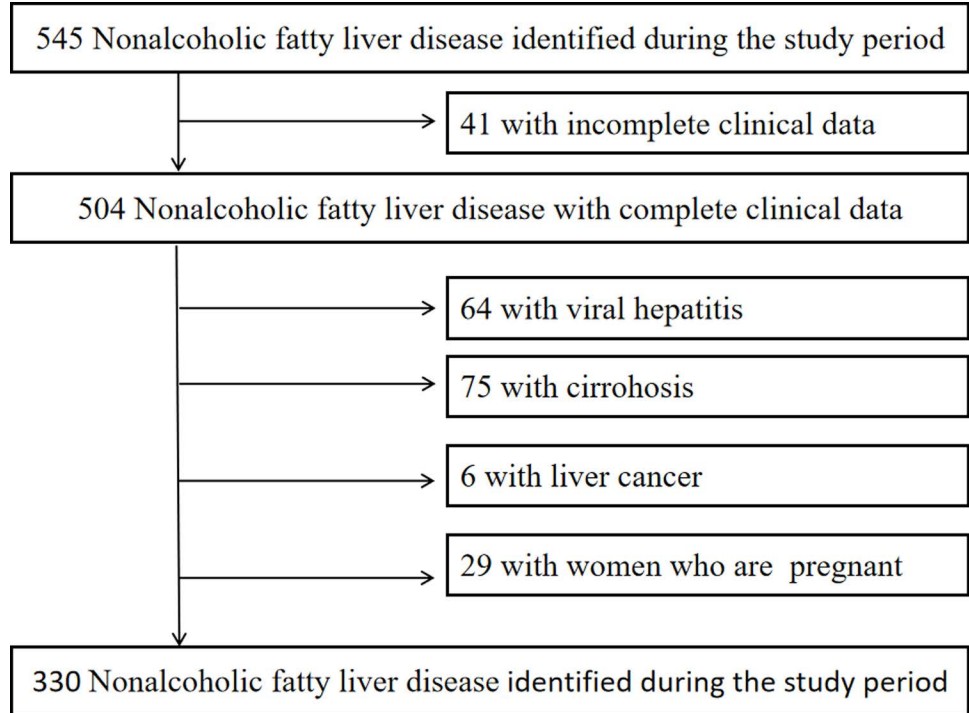

**Fig. 1. Flow diagram of study selection.**

different points in the intercostal region.The average value was calculated as the final output, with CAP in dB/m. ≤ 238 CAP < 259 was considered mild fatty liver, 259 ≤ CAP < 292 was considered moderate fatty liver, and CAP ≥ 292 was considered severe fatty liver.

## 2.2 Methods of collecting patient data

With the informed consent of the study subjects, the general clinical data of NAFLD patients were measured using uniform standards; body fat analysis was conducted by the Department of Hepatology, Shuguang Hospital, Shanghai University of Traditional Chinese Medicine, with the instrument of Inbody 720 Body Composition Analyzer, and VFA were measured.

**2.2.2 Biochemical Index Testing.** With the informed consent of the study subjects, the Laboratory Department of Shuguang Hospital, affiliated with Shanghai University of Traditional Chinese Medicine, used the Beckman AU5800 automatic biochemical analyzer for testing, including ALT, AST, ALP, GGT, TG, TC, LDL, HDL and other indicators;

**2.2.3 Liver elasticity ultrasound examination.** With the informed consent of the study subjects, the elastic ultrasound CAP values were examined in the ultrasound room of Shuguang Hospital, affiliated with the Shanghai University of Traditional Chinese Medicine, with the instrument Echosens Fibroscan-520.

## 2.3 Statistical analysis methods

Data were statistically analyzed using IBM SPSS 26.0 software, and results of independent factors were visualized and plotted using R software (version 4.3.3). Continuous variables that fit a normal distribution were expressed as mean ± standard deviation. Continuous variables that do not fit the normal distribution were expressed as median (first quartile, third quartile). Categorical information between groups was compared using the chi-square test, and the remaining continuous variables were compared using the Kruskal-Wallis H test. When there was a statistical difference in the results of the descriptive analysis, multivariate ordinal logistic regression was performed on the factor. One-way logistic regression analysis was performed for indicators with statistically different results in descriptive statistics. Multiple logistic regression analysis was performed for indicators that differed from the one-way logistic regression analysis results. The male was used as the reference sex. Logistic regression analysis was performed for age, ALT, AST, ALP, GGT, TC, TG, HDL, LDL, FBG, and VFA. $P < 0.05$ represents statistical significance.

## 3. Result

### 3.1 Comparison of the three groups of general data and biochemical indicators

Based on the above criteria, 330 samples were collected, of which 173 were males and 157 were females. Of the 330 samples, 36 were patients with mild NAFLD, 81 had moderate NAFLD, and 213 had severe NAFLD. There were statistically significant differences in age (AGE), alanine aminotransferase (ALT), aspartate aminotransferase (AST), alkaline phosphatase (ALP), gamma-glutamyl transferase (GGT), triglycerides (TG), fasting blood glucose (FBG), and visceral fat area (VFA) among the three groups (P < 0.05). However, differences in gender, total cholesterol (TC), low-density lipoprotein (LDL), and high-density lipoprotein (HDL) were not statistically significant (Table 1). All patients were stratified and analyzed by gender. Firstly, 173 males were classified into mild, moderate, and severe fatty liver, with 19 patients with mild NAFLD, 33 patients with moderate NAFLD, and 121 patients with severe NAFLD. The differences in AGE, ALT, AST, GGT, TG, and VFA were statistically significant among the three groups(P < 0.05). However, differences in ALP, TC, LDL, HDL, and FBG differences were not statistically significant (Table 2). The 157 females were categorized into mild, moderate, and severe fatty liver, with 17 patients with mild NAFLD, 48 patients with moderate NAFLD, and 92 patients with severe NAFLD. The differences in ALT, ALP, GGT,

TG, FBG, and VFA among the three groups were statistically significant(P < 0.05). However, differences in AGE, AST, TC, LDL, and HDL were not statistically significant (Table 3). All patients were stratified and analyzed by age, and all 330 patients were divided into two groups, namely the young and middle-aged group and the elderly group. Among them, 204 young and middle-aged NAFLD patients (age < 60 years) were categorized into mild, moderate and severe fatty liver, including 19 patients with mild NAFLD, 43 patients with moderate NAFLD and 142 patients with severe NAFLD. Statistically significant differences in age (AGE), ALT, AST, GGT, TG, and VFA were found between the three groups (P < 0.05). However, there were no statistically significant differences in gender, ALP, TC, LDL, HDL, and FBG (Table 4).126 elderly NAFLD patients (age > 60 years) were categorized as having mild, moderate, and severe fatty livers, including 17 patients with mild NAFLD, 38 patients with moderate NAFLD, and 71 patients with severe NAFLD. Statistically significant differences in age (AGE), ALT, ALP, GGT, TG, FBG, and VFA were found between the three groups (P < 0.05). However, there were no statistically significant differences in gender, AST, TC, LDL and HDL (Table 5).

Table 1. General information and biochemical test results in the total population of three groups of NAFLD patients.

| Characteristic | Mild NAFLD (n = 36) | Moderate NAFLD (n = 81) | Severe NAFLD (n = 213) | X²/H | P |
|---|---|---|---|---|---|
| Gender,n(%) | | | | 7.000 | 0.321 |
| Female | 17(5.2%) | 48(14.5%) | 92(27.9%) | | |
| Male | 19(5.7%) | 33(10.0%) | 121(36.7%) | | |
| Age(years) | 48.50(39.25,64.00) | 59.00(51.00,64.00) | 50.00(37.00,63.00) | 8.053 | 0.018 |
| ALT/(U.L$^{-1}$) | 20.00(12.25,25.79) | 24.00(16.00,40.50) | 32.00(21.00,56.00) | 28.792 | <0.001 |
| AST/(U.L$^{-1}$) | 22.50(20.00,31.00) | 24.00(21.00,33.50) | 29.00(22.00,38.50) | 15.333 | <0.001 |
| ALP/(U.L$^{-1}$) | 77.50(67.00,91.50) | 88.00(75.00,104.00) | 89.00(78.00,107.00) | 11.309 | 0.004 |
| GGT/(U.L$^{-1}$) | 21.67(26.34,33.07) | 30.62(19.49,51.26) | 32.86(24.86,57.03) | 17.354 | <0.001 |
| TG/(mmol.L$^{-1}$) | 1.38(0.99,2.29) | 1.84(1.36,2.82) | 2.25(1.59,3.27) | 20.009 | <0.001 |
| TC/(mmol.L$^{-1}$) | 4.92(4.39,5.87) | 5.19(4.58,5.81) | 5.17(4.53,5.88) | 0.132 | 0.936 |
| LDL/(mmol.L$^{-1}$) | 2.78(2.31,3.52) | 3.01(2.48,3.40) | 2.92(2.47,3.53) | 0.141 | 0.932 |
| HDL/(mmol.L$^{-1}$) | 1.25(1.09,1.43) | 1.19(1.08,1.39) | 1.16(1.01,1.30) | 4.125 | 0.127 |
| FBG/(mmol.L$^{-1}$) | 4.94(4.64,5.24) | 5.33(4.86,6.16) | 5.49(5.00,6.51) | 16.676 | <0.001 |
| VFA(cm²) | 104.85(89.63,116.42) | 107.20(90.65,129.95) | 123.20(100.25,152.20) | 19.469 | <0.001 |

Table 2. Male patients information and biochemical detection indexes of the three groups.

| Characteristic | Mild NAFLD (n = 19) | Moderate NAFLD (n = 33) | Severe NAFLD (n = 121) | X²/H | P |
|---|---|---|---|---|---|
| Age(years) | 49.00(40.00,66.00) | 55.00(41.50,63.50) | 42.00(34.00,57.50) | 10.197 | 0.006 |
| ALT/(U.L$^{-1}$) | 21.00(12.00,32.00) | 26.00(19.50,48.50) | 37.00(26.50,66.00) | 16.599 | <0.001 |
| AST/(U.L$^{-1}$) | 25.46(20.00,31.00) | 25.00(21.00,39.00) | 31.00(23.50,41.50) | 8.601 | 0.014 |
| ALP/(U.L$^{-1}$) | 80.00(73.00,92.00) | 90.00(72.00,103.50) | 86.00(75.00,99.50) | 2.556 | 0.279 |
| GGT/(U.L$^{-1}$) | 28.68(20.95,41.64) | 37.39(23.61,57.56) | 43.82(28.08,61.51) | 7.170 | 0.028 |
| TG/(mmol.L$^{-1}$) | 1.72(1.01,2.48) | 1.84(1.49,2.71) | 2.25(1.65,3.25) | 6.317 | 0.042 |
| TC/(mmol.L$^{-1}$) | 4.59(4.03,5.42) | 4.93(4.39,5.66) | 4.96(4.37,5.44) | 0.880 | 0.644 |
| LDL/(mmol.L$^{-1}$) | 2.58(2.11,3.18) | 2.96(2.39,3.47) | 2.77(2.41,3.24) | 2.202 | 0.332 |
| HDL/(mmol.L$^{-1}$) | 1.16(0.92,1.26) | 1.16(0.99,1.27) | 1.08(0.98,1.25) | 0.750 | 0.687 |
| FBG/(mmol.L$^{-1}$) | 4.94(4.62,5.64) | 5.29(4.85,6.68) | 5.31(4.95,6.23) | 4.391 | 0.111 |
| VFA(cm²) | 94.90(74.40,109.20) | 92.30(84.60,110.70) | 111.40(96.40,139.95) | 19.098 | <0.001 |

## 3.2 Logistic regression analysis results

In the total population, indicators with statistically different statistics were analyzed by univariate logistic regression, and AGE, ALT, AST, ALP, GGT, TG, FBG, and VFA were statistically significant enough to be included in the multifactorial logistic regression analysis, and the final multifactorial logistic regression results showed that ALT, FBG, TG, and VFA were independent risk for progression of NAFLD factors($P < 0.05$). The above statistically significant indicators of male patients were included in the univariate logistic regression to analyze the results of different descriptive statistics; AGE, ALT, AST, TG, and VFA were statistically significant, and GGT was not statistically significant. The results that were significant in the univariate analysis were included in the multifactorial logistic regression analysis, and the final multifactorial logistic regression results showed that ALT, TG, and VFA were independent risk factors for disease progression in NAFLD in the male patient population ($P < 0.05$). The above statistically significant indicators of female patients were included in the one-way logistic regression to analyze the different descriptive statistical results; ALP, GGT, TG, FBG, and VFA were statistically significant, and ALT was not

**Table 3. Female patients information and biochemical detection indexes of the three groups.**

| Characteristic | Mild NAFLD (n=17) | Moderate NAFLD (n=48) | Severe NAFLD (n=92) | X²/H | P |
|---|---|---|---|---|---|
| Age(years) | 48.00(35.00,62.00) | 60.00(54.00,64.00) | 58.50(44.00,65.00) | 3.106 | 0.212 |
| ALT/(U.L⁻¹) | 18.00(12.00,23.50) | 22.00(13.25,29.75) | 24.00(17.25,41.25) | 9.909 | 0.007 |
| AST/(U.L⁻¹) | 20.00(19.00,28.50) | 24.00(21.00,29.00) | 26.50(21.00,34.75) | 5.673 | 0.059 |
| ALP/(U.L⁻¹) | 74.00(61.00,88.00) | 85.00(75.25,104.50) | 94.00(84.00,111.00) | 11.902 | 0.003 |
| GGT/(U.L⁻¹) | 17.42(13.46,24.12) | 22.21(17.49,42.43) | 26.82(19.64,45.19) | 10.467 | 0.005 |
| TG/(mmol.L⁻¹) | 1.34(0.94,1.94) | 1.86(1.25,2.91) | 2.25(1.53,3.29) | 15.182 | 0.001 |
| TC/(mmol.L⁻¹) | 5.59(4.60,6.99) | 5.31(4.60,5.90) | 5.54(4.90,6.36) | 1.718 | 0.424 |
| LDL/(mmol.L⁻¹) | 3.46(2.58,4.72) | 3.04(2.50,3.40) | 3.25(2.65,3.77) | 2.108 | 0.348 |
| HDL/(mmol.L⁻¹) | 1.38(1.25,1.52) | 1.24(1.10,1.53) | 1.25(1.13,1.41) | 3.511 | 0.173 |
| FBG/(mmol.L⁻¹) | 4.94(4.59.5.24) | 5.38(4.87,6.05) | 5.88(5.11,6.85) | 17.113 | <0.001 |
| VFA(cm²) | 108.00(98.30,125.75) | 118.30(99.68,134.13) | 133.50(107.85,165.78) | 10.182 | 0.006 |

**Table 4. Young and middle-aged patients information and biochemical detection indexes of the three groups.**

| Characteristic | Mild NAFLD (n=19) | Moderate NAFLD (n=43) | Severe NAFLD (n=142) | X²/H | P |
|---|---|---|---|---|---|
| Gender,n(%) | | | | 5.950 | 0.510 |
| Female | 9(4.4%) | 23(11.3%) | 48(23.5%) | | |
| Male | 10(4.9%) | 20(9.8%) | 94(46.1%) | | |
| Age(years) | 40.00(35.00,43.00) | 52.00(40.00,55.00) | 39.50(33.00,50.00) | 14.946 | 0.001 |
| ALT/(U.L⁻¹) | 22.00(13.00,33.00) | 25.00(16.00,48.00) | 34.00(23.57,60.50) | 13.397 | 0.001 |
| AST/(U.L⁻¹) | 21.00(19.00,31.00) | 24.00(21.00,40.00) | 30.50(23.00,40.00) | 10.771 | 0.005 |
| ALP/(U.L⁻¹) | 77.00(70.00,90.00) | 90.00(74.00,105.00) | 87.50(78.00,105.25) | 4.867 | 0.088 |
| GGT/(U.L⁻¹) | 26.17(15.58,40.91) | 26.67(20.92,54.15) | 42.45(26.87,61.00) | 12.922 | 0.002 |
| TG/(mmol.L⁻¹) | 1.37(1.17,2.43) | 2.06(1.41,2.85) | 2.21(1.60,3.18) | 10.355 | 0.006 |
| TC/(mmol.L⁻¹) | 4.71(4.38,5.59) | 5.08(4.48,5.81) | 5.05(4.46,5.66) | 0.532 | 0.767 |
| LDL/(mmol.L⁻¹) | 2.61(2.42,3.46) | 2.99(2.37,3.40) | 2.89(2.46,3.40) | 0.276 | 0.871 |
| HDL/(mmol.L⁻¹) | 1.19(1.09,1.28) | 1.18(1.00,1.38) | 1.16(1.00,1.27) | 1.492 | 0.474 |
| FBG/(mmol.L⁻¹) | 5.02(4.77,5.64) | 5.29(4.86,5.85) | 5.36(4.98,6.26) | 3.014 | 0.222 |
| VFA(cm²) | 107.90(79.80,117.80) | 105.00(89.30,129.90) | 121.00(99.80,155.73) | 11.265 | 0.004 |

statistically significant. The results that were significant in the univariate analysis were included in the multivariate logistic regression analysis, and the final multivariate logistic regression results showed that FBG, TG, and VFA were independent risk factors for disease progression in NAFLD in the female patient population(P < 0.05). The above statistically significant indicators of young and middle-aged patients were included in the one-way logistic regression analysis of different descriptive statistics results for ALT, AST, GGT, TG, and VFA which were statistically significant, and AGE which was not statistically significant. The results that were significant in the univariate analysis were included in the multifactorial logistic regression analysis, and the final multifactorial logistic regression results showed that ALT, TG, and VFA were independent risk factors for disease progression in NAFLD in the young and middle-aged patient population (P < 0.05). The above statistically significant indicators of elderly patients were included in the one-way logistic regression analysis of different descriptive statistics results of indicators, AGE, ALT, ALP, TG, FBG, and VFA which were statistically significant. The results of the one-way analysis of the meaningful results were included in the multifactorial logistic regression analysis. The final multifactorial logistic in the group of elderly patients regression results showed that AGE, ALT, TG, and FBG were independent risk factors for disease progression in NAFLD (Table 6; Fig 2).

## 4. Discussion

In this study, we analyzed data from 330 patients to investigate the risk factors associated with causing NAFLD disease progression. ALT, FBG, TG, and VFA were independent risk factors for the progression of NAFLD in the whole population.

Among the identified factors, ALT is a standard indicator of liver function. It was found to be an independent risk factor for disease progression in NAFLD. For every one-unit increase in ALT, the likelihood of exacerbating NAFLD tendency was elevated by 2.8%. Previous findings have shown a correlation between ALT and the severity of NAFLD [13,14], Elevated serum ALT is associated with hepatic oxidative stress and to some extent reflects the severity of hepatic inflammation [15], and our findings are consistent with previous research, which suggests that ALT can predict and screen for severe NAFLD to a certain extent. Similarly, FBG was an independent risk factor, with each unit increase raising the likelihood of progression by 34.5%, consistent with evidence linking hyperglycemia to increased hepatic lipid accumulation.

**Table 5. Elderly patients information and biochemical detection indexes of the three groups.**

| Characteristic | Mild NAFLD (n = 17) | Moderate NAFLD  (n = 38) | Severe NAFLD (n = 71) | $X^2$/H | P |
|---|---|---|---|---|---|
| Gender,n(%) | | | | 0.785 | 0.410 |
| Female | 8(6.4%) | 25(19.8%) | 44(35.0%) | | |
| Male | 9(7.1%) | 13(10.3%) | 27(21.4%) | | |
| Age(years) | 64.00(62.00,66.50) | 64.00(60.75,65.25) | 65.00(63.00,67.00) | 7.002 | 0.030 |
| ALT/(U.L$^{-1}$) | 19.00(11.00,23.58) | 24.00(15.75,30.50) | 26.00(19.00,42.00) | 11.973 | 0.003 |
| AST/(U.L$^{-1}$) | 25.00(20.00,31.00) | 24.00(20.75,28.25) | 25.00(22.00,34.00) | 3.489 | 0.175 |
| ALP/(U.L$^{-1}$) | 78.00(58.50,95.50) | 86.00(75.75,98.50) | 93.00(77.00,109.00) | 6.750 | 0.034 |
| GGT/(U.L$^{-1}$) | 20.95(14.61,30.01) | 31.35(19.00,48.93) | 26.26(22.01,39.16) | 5.092 | 0.078 |
| TG/(mmol.L$^{-1}$) | 1.70(0.94,2.16) | 1.81(1.29,2.54) | 2.27(1.57,3.41) | 10.308 | 0.006 |
| TC/(mmol.L$^{-1}$) | 5.41(4.45,6.12) | 5.26(4.70,5.78) | 5.47(4.75,6.31) | 0.975 | 0.614 |
| LDL/(mmol.L$^{-1}$) | 3.12(2.28,3.73) | 3.03(2.52,3.42) | 3.05(2.50,3.76) | 0.077 | 0.962 |
| HDL/(mmol.L$^{-1}$) | 1.38(1.11,1.52) | 1.21(1.11,1.40) | 1.22(1.06,1.41) | 1.919 | 0.383 |
| FBG/(mmol.L$^{-1}$) | 4.76(4.45,5.09) | 5.38(4.86,6.58) | 5.87(5.08,6.90) | 17.167 | <0.001 |
| VFA(cm$^2$) | 98.70(90.85,116.35) | 111.00(92.28,130.28) | 124.20(103.80,148.10) | 8.385 | 0.015 |

Younossi ZM [16] et al., in a meta-analysis of 80 studies from 20 countries, found that 55.5% of 49,419 patients with T2DM suffered from NAFLD. It has also been shown that blood glucose is elevated, which leads to a further increase in the concentration of lipids in the liver [17], and that patients with T2DM are at a significantly increased risk of developing severe liver disease in the future [18]. Furthermore, TG was the strongest independent risk factor. Logistic regression

Table 6. Risk Factors for Developing Non-Alcoholic Fatty Liver Disease from Sex Stratification and Age Stratification.

| Characteristic | Univariate analysis | | Multivariate analysis | |
|---|---|---|---|---|
| | Odds ratio(95% CI) | P-value | Odds ratio(95% CI) | P-value |
| Total | | | | |
| Age | 0.795(0.677-0.942) | 0.007 | 0.974(0.948-1.000) | 0.071 |
| ALT | 1.022(1.011-1.034) | <0.001 | 1.028(1.010-1.046) | 0.002 |
| AST | 1.024(1.007-1.042) | 0.006 | 0.978(0.951-1.006) | 0.120 |
| ALP | 1.014(1.004-1.024) | 0.006 | 1.007(0.996-1.018) | 0.230 |
| GGT | 1.008(1.001-1.015) | 0.001 | 1.000(0.992-1.007) | 0.901 |
| TG | 1.493(1.206-1.848) | <0.001 | 1.458(1.164-1.826) | 0.001 |
| FBG | 1.343(1.121-1.610) | 0.001 | 1.347(1.100-1.650) | 0.004 |
| VFA | 1.015(1.008-1.022) | <0.001 | 1.015(1.007-1.023) | <0.001 |
| Male | | | | |
| Age | 0.964(0.941-0.987) | 0.002 | 0.977(0.951-1.004) | 0.053 |
| ALT | 1.026(1.105-1.043) | 0.001 | 1.026(1.002-1.051) | 0.035 |
| AST | 1.024(1.000-1.049) | 0.049 | 0.980(0.947-1.014) | 0.249 |
| GGT | 1.004(0.996-1.012) | 0.324 | | |
| TG | 1.438(1.059-1.952) | 0.020 | 1.459(1.056-2.016) | 0.022 |
| VFA | 1.024(1.011-1.037) | <0.001 | 1.022(1.009-1.037) | 0.001 |
| Female | | | | |
| ALT | 1.015(0.999-1.031) | 0.064 | | |
| ALP | 1.025(1.010-1.040) | 0.001 | 1.012(0.996-1.028) | 0.134 |
| GGT | 1.017(1.001-1.034) | 0.040 | 1.008(0.992-1.025) | 0.324 |
| TG | 1.542(1.143-2.077) | 0.005 | 1.404(1.021-1.931) | 0.037 |
| FBG | 1.946(1.373-2.759) | <0.001 | 1.861(1.306-2.718) | 0.001 |
| VFA | 1.014(1.005-1.024) | 0.003 | 1.015(1.005-1.027) | 0.005 |
| Young and middle-aged | | | | |
| Age | 0.973(0.946-0.999) | 0.062 | | |
| ALT | 1.016(1.004-1.028) | 0.001 | 1.026(1.010-1.048) | 0.010 |
| AST | 1.018(0.999-1.039) | 0.007 | 0.969(0.936-1.003) | 0.072 |
| GGT | 1.010(1.000-1.020) | 0.045 | 1.006(0.996-1.017) | 0.249 |
| TG | 1.495(1.108-2.014) | 0.008 | 1.523(1.100-2.119) | 0.011 |
| VFA | 1.016(1.007-1.025) | 0.001 | 1.018(1.008-1.028) | <0.001 |
| Elderly | | | | |
| Age | 1.141(1.013-1.285) | 0.003 | 1.157(1.012-1.323) | 0.032 |
| ALT | 1.036(1.011-1.060) | 0.004 | 1.030(1.004-1.058) | 0.022 |
| ALP | 1.020(1.005-1.037) | 0.011 | 1.013(0.996-1.032) | 0.137 |
| TG | 1.495(1.105-2.104) | 0.009 | 1.418(1.041-1.931) | 0.027 |
| FBG | 1.429(1.112-1.837) | 0.005 | 1.389(1.052-1.833) | 0.020 |
| VFA | 1.014(1.003-1.025) | 0.012 | 1.010(0.998-1.022) | 0.098 |

analysis revealed that each unit increase in TG was associated with a 45.8% higher likelihood of NAFLD progression. This aligns with prior studies suggesting that hypertriglyceridemia exacerbates insulin resistance [19], which in turn promotes NAFLD by inducing lipolysis of adipose tissue triglycerides and de novo synthesis of triglycerides in the liver [20]. Previous studies have shown that visceral obesity is strongly associated with NAFLD and visceral fat accumulation is a clinical predictor of NAFLD [21,22]. However, limited research has explored the role of visceral fat area (VFA) as a predictor of NAFLD progression. The findings of this study demonstrate that VFA is positively associated with the severity of NAFLD and serves as an independent risk factor for its progression.

Based on further analysis, the following conclusions were drawn. First, multifactorial logistic regression analyses demonstrated that TG was the most significant risk factor, maintaining its dominance across all population groups, consistent with findings from previous studies [23,24]. Visceral obesity leads to systemic inflammation and insulin resistance, which promotes energy flow to the liver and increases metabolic stress on hepatocytes. When hepatocytes fail to adapt, this metabolic stress triggers oxidative damage or direct lipotoxicity from free fatty acids. This process leads to adipocyte apoptosis, sterile inflammation, and chronic inflammation, which in turn promote liver fibrosis, cirrhosis, and tumorigenesis [25]. Through sex- and age-stratified analyses we found that VFA was an independent risk factor for the progression of NAFLD in males, females, and young and middle-aged adults, whereas it was not in the older age groups. This is entirely consistent with a previous study in which the cumulative incidence of NAFLD increased with VFA in both men and women, and further analyses showed that an increase in VFA especially in those younger than 60 years of age led to an increase

| Characteristic | adjusted OR(95% CI) | | P value |
|---|---|---|---|
| Total | | | |
| ALT | 1.028(1.010-1.046) | | 0.002 |
| TG | 1.458(1.164-1.826) | | 0.001 |
| FBG | 1.347(1.100-1.650) | | 0.004 |
| VFA | 1.015(1.007-1.023) | | <0.001 |
| Male | | | |
| ALT | 1.026(1.002-1.051) | | 0.035 |
| TG | 1.459(1.056-2.016) | | 0.022 |
| VFA | 1.022(1.009-1.037) | | 0.001 |
| Female | | | |
| TG | 1.404(1.021-1.931) | | 0.037 |
| FBG | 1.861(1.306-2.718) | | 0.001 |
| VFA | 1.015(1.005-1.027) | | 0.005 |
| Young and middle-aged | | | |
| ALT | 1.026(1.010-1.048) | | 0.010 |
| TG | 1.523(1.100-2.119) | | 0.011 |
| VFA | 1.018(1.008-1.028) | | <0.001 |
| Elderly | | | |
| Age | 1.157(1.012-1.323) | | 0.032 |
| ALT | 1.030(1.004-1.058) | | 0.022 |
| TG | 1.418(1.041-1.931) | | 0.027 |
| FBG | 1.389(1.052-1.833) | | 0.020 |

**Fig. 2. Forest plots stratified for age and sex by multifactorial logistic regression analysis are indicators related to risk factors influencing the progression of NAFLD.**

in NAFLD incidence. In contrast, these differences disappeared in those over 60 years of age [26]. Fat in the visceral region also has a higher rate of lipolysis and pro-inflammatory adipokine production, thereby exposing the liver to higher concentrations of fatty acids [27], which can lead to NAFLD progression. In the female population, fasting blood glucose (FBG) emerged as the strongest independent risk factor, with the likelihood of NAFLD progression increasing by 86.1% for each unit increase in FBG. This suggests that the female NAFLD patients with hyperglycemia in the clinic need fasting blood glucose control while actively treating to prevent the continued progression of NAFLD. In the elderly population, age is an independent risk factor for NAFLD disease progression, which is consistent with some studies in recent years [24,28]. This is because the molecular mechanisms by which aging leads to hepatocellular damage include increased ROS formation and DNA damage [29,30], which aggravates NAFLD [31]. In addition, we found by stratified analysis that ALT was a risk factor for NAFLD progression in the male, young middle-aged, and elderly populations, but not in the female population. This may be because estrogen may have a favorable effect on hepatic lipid metabolism [32], reducing ALT elevation to some extent.

In summary, our analysis of patients diagnosed with NAFLD at Shanghai Shuguang Hospital revealed that ALT, TG, FBG, and VFA are closely associated with disease severity. Physicians should monitor these factors to prevent NAFLD. In males, ALT, TG, and VFA should be prioritized for monitoring. In females, FBG, TG, and VFA warrant closer attention. For younger and middle-aged populations, ALT, TG, and VFA are critical, while in elderly patients, ALT, TG, and FBG should be the focus. Regular medical check-ups and early interventions are essential, particularly for elderly patients, to prevent progression to cirrhosis or hepatocellular carcinoma. This study has several strengths. First, the inclusion of pre-pandemic data avoided potential confounding effects of the COVID-19 pandemic on metabolic health. Second, the diagnostic accuracy was enhanced by using liver ultrasound and FibroTouch elastography for grading NAFLD severity. However, there are notable limitations. Firstly, this study was a retrospective, single-center, cross-sectional study with some selection bias. Secondly, the prediction model did not include some potential risk factors, such as diet and smoking history. Lastly, the sample size included in our study was small. Therefore, more multicenter, long-term follow-up studies with large sample sizes are needed for external validation of further studies.

## 5. Conclusion

Our study identified several risk factors for NAFLD progression, highlighting the significance of considering gender and age in its development. This study emphasizes the need for tailored risk assessment, prevention, and treatment strategies by identifying high-risk populations. Enhanced monitoring of key risk factors is crucial for implementing effective population-based interventions against NAFLD.

## Author contributions

**Conceptualization:** Junran Yang.

**Data curation:** Junran Yang.

**Formal analysis:** Junran Yang.

**Funding acquisition:** Zhenhua Zhou.

**Methodology:** Zhenhua Zhou.

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
