## [Decision Letter · Decision Letter 0]

14 Jan 2025

PONE-D-24-34142Analysis and study of risk factors related to the progression of non-alcoholic fatty liver diseasePLOS ONE

Dear Dr. Zhou,

Thank you for submitting your manuscript to PLOS ONE. After careful consideration, we feel that it has merit but does not fully meet PLOS ONE’s publication criteria as it currently stands. Therefore, we invite you to submit a revised version of the manuscript that addresses the points raised during the review process.

**ACADEMIC EDITOR:**

This retrospective cohort study represents a valuable contribution to understanding the risk factors associated with NAFLD progression. The study benefits from several methodological strengths, including data collection from before the COVID-19 pandemic, thereby avoiding potential confounding effects, and the use of both ultrasound and FibroTouch elastography for accurate diagnosis. The stratified analysis by gender and age provides important insights into population-specific risk factors.

However, several aspects of the manuscript require attention before publication. The statistical methodology needs more rigorous documentation, particularly regarding sample size calculation and power analysis. The Results section would benefit from more consistent presentation of statistical findings, with standardized reporting of significance levels and effect sizes. Some contradictions appear in the Discussion section, particularly regarding the relationship between age, visceral fat area (VFA), and NAFLD progression in elderly populations, which requires clarification.

The manuscript would be substantially strengthened by a more comprehensive integration of recent literature, particularly in comparing findings with similar studies. The discussion of potential mechanisms underlying the observed associations could be expanded to provide better context for the findings. Some technical issues need addressing, including standardization of abbreviations (e.g., FBS vs. FBG) and professional English language editing to improve clarity and precision.

Despite these limitations, the study provides valuable insights into the differential risk factors for NAFLD progression across demographic groups. With appropriate revision to address these concerns, particularly the resolution of contradictory findings and strengthening of the statistical presentation, this manuscript could make a meaningful contribution to the field.

We look forward to receiving your revised manuscript.

Kind regards,

Jincheng Wang

Academic Editor

PLOS ONE

Journal Requirements:

“Anhui Provincial Natural Science Foundation(2023085MH293)

Shanghai Municipal Science and Technology Commission Science and Technology Innovation Action Plan Science and Technology Support Project(NO.21S1900400)

Major scientific research project in colleges and universities in Anhui Province(NO.2023AH040098)”

“Anhui Provincial Natural Science Foundation(2023085MH293)

Shanghai Municipal Science and Technology Commission Science and Technology Innovation Action Plan Science and Technology Support Project(NO.21S1900400)

Major scientific research project in colleges and universities in Anhui Province(NO.2023AH040098)”

3. Please include captions for your Supporting Information files at the end of your manuscript, and update any in-text citations to match accordingly. Please see our Supporting Information guidelines for more information: http://journals.plos.org/plosone/s/supporting-information .

Additional Editor Comments (if provided):

This retrospective cohort study represents a valuable contribution to understanding the risk factors associated with NAFLD progression. The study benefits from several methodological strengths, including data collection from before the COVID-19 pandemic, thereby avoiding potential confounding effects, and the use of both ultrasound and FibroTouch elastography for accurate diagnosis. The stratified analysis by gender and age provides important insights into population-specific risk factors.

However, several aspects of the manuscript require attention before publication. The statistical methodology needs more rigorous documentation, particularly regarding sample size calculation and power analysis. The Results section would benefit from more consistent presentation of statistical findings, with standardized reporting of significance levels and effect sizes. Some contradictions appear in the Discussion section, particularly regarding the relationship between age, visceral fat area (VFA), and NAFLD progression in elderly populations, which requires clarification.

The manuscript would be substantially strengthened by a more comprehensive integration of recent literature, particularly in comparing findings with similar studies. The discussion of potential mechanisms underlying the observed associations could be expanded to provide better context for the findings. Some technical issues need addressing, including standardization of abbreviations (e.g., FBS vs. FBG) and professional English language editing to improve clarity and precision.

Despite these limitations, the study provides valuable insights into the differential risk factors for NAFLD progression across demographic groups. With appropriate revision to address these concerns, particularly the resolution of contradictory findings and strengthening of the statistical presentation, this manuscript could make a meaningful contribution to the field.

Reviewers' comments:

Reviewer's Responses to Questions

**Comments to the Author**

1. Is the manuscript technically sound, and do the data support the conclusions?

Reviewer #1: Yes

Reviewer #2: Partly

Reviewer #3: Yes

2. Has the statistical analysis been performed appropriately and rigorously?

Reviewer #1: Yes

Reviewer #2: I Don't Know

Reviewer #3: Yes

3. Have the authors made all data underlying the findings in their manuscript fully available?

Reviewer #1: Yes

Reviewer #2: Yes

Reviewer #3: Yes

4. Is the manuscript presented in an intelligible fashion and written in standard English?

Reviewer #1: Yes

Reviewer #2: No

Reviewer #3: Yes

5. Review Comments to the Author

Reviewer #1: This study was a well-controlled study. Statistical procedures seemed steady. The potential confounders and occupational exposure factors were excluded to conduct their results. These are the strengths of their study. However, the following comments should also be considered:

1) Result section in abstract is not clear. Please revise this section.

2) How sample size was calculated?

3) A language editing is recommended by a native speaker to revise some grammatical errors.

4) Please update some previous references with new references.

Reviewer #2: Dear Authors,

Thank you for submitting your manuscript titled “Analysis and study of risk factors related to the progression of non-alcoholic fatty liver disease “for consideration. I appreciate the effort and dedication that has gone into your research. After a thorough review, I have provided detailed feedback and suggestions that I believe will enhance the clarity and impact of your work. I encourage you to address these points to strengthen your manuscript further.

1. On page 13, Part 3 (Results), Section 3.1 (Comparison of general data and biochemical indicators of the three groups), the first paragraph stating "There were statistically significant differences…….. (Table 1; P < 0.05)" requires a fundamental revision. Additionally, the placement of abbreviated words needs correction, such as using 'FBS' for blood lipids, which is incorrect, and using 'TC' as an abbreviation for blood lipids (Do you mean Total Cholesterol? and GGt is Gamma-Glutamyl Transferase,…

2. In the Results section, it's recommended to place the significance level (P < 0.05) at the end of sentences discussing significant results, rather than following those addressing non-significant findings.

3. In table 1, please add the Age unit in the table.

4. There is an inconsistency regarding ALP in the explanation. According to Table 5, with a p-value of 0.034, ALP is significant at the conventional level of 0.05. However, it is incorrectly stated as not significant for individuals over 60 years. This needs correction to reflect the data accurately.

5. Please use consistent abbreviations throughout the text for a specific topic. For instance, you have used FBS throughout the article, but FBG once at the beginning of the discussion. Using different terms confuses the readers.

6. These two parts in the article's discussion section about elderly people appear to be contradictory. Please review them carefully and make corrections as needed:

“This is entirely consistent with a previous study in which the cumulative incidence of NAFLD increased with VFA in both men and women, and further analyses showed that an increase in VFA especially in those younger than 60 years of age led to an increase in NAFLD incidence. In contrast, these differences disappeared in those over 60 years of age.”

And,

“In the elderly population, age is an independent risk factor for NAFLD disease progression, which is consistent with some studies in recent years. This may be because aging is accompanied by abdominal obesity and excess visceral fat, which leads to an increase in insulin resistance and pro-inflammatory cytokine secretion, and the accumulation of excess fat in the liver and the molecular mechanisms of hepatocellular damage caused by aging include increased ROS formation, DNA damage, which aggravates NAFLD.”

7. A more effective comparison with similar studies could be incorporated into the discussion section of the article, potentially drawing on the findings from other relevant research. For example, the results of the following article can be helpful in the discussion section:

“Metabolic risk factors and incident advanced liver disease in non-alcoholic fatty liver disease (NAFLD): A systematic review and meta-analysis of population-based observational studies,2020”

8. There are few typographical or grammatical errors that should be corrected. Please, check.

Best Regards

Reviewer #3: The data and results of the article"Analysis and study of risk factors related to the progression of non-alcoholic fatty liver disease" are not valuable for publishing in this journal, considering the population, indicators, duration of data collection, and the journal's standards.

6. PLOS authors have the option to publish the peer review history of their article (what does this mean? ). If published, this will include your full peer review and any attached files.

**Do you want your identity to be public for this peer review?** For information about this choice, including consent withdrawal, please see our Privacy Policy .

Reviewer #1: **Yes: ** Helda Tutunchi

Reviewer #2: **Yes: ** Roshanak Ghods

Reviewer #3: No

---

## [Author Response · Author response to Decision Letter 0]

11 Feb 2025

Response to Reviewers

Dear editors and reviewers,

We sincerely appreciate your constructive comments and suggestions on our manuscript, titled "Analysis and Study of Risk Factors Related to the Progression of Non-Alcoholic Fatty Liver Disease: A Retrospective Cohort Study" (ID: PONE-D-24-34142). Your feedback has been invaluable in improving the quality and clarity of our manuscript.

In the revised manuscript, we have made significant updates based on your insights. Below, we provide detailed responses to each comment. We hope these revisions meet your expectations and address all concern

The main revisions in the new manuscript are:

1. We ensured consistency in the use of abbreviations, resolving discrepancies such as "FBS" vs. "FBG."

2. Recent literature has been integrated, and an extended discussion of potential mechanisms underlying the observed associations has been included.

3. For sample size calculation and efficacy analysis, we have provided the sample size calculation formula to illustrate the reasonableness of our sample size, and I will clarify the specific formula in my reply to reviewer 1.

4. Contradictions in the discussion section, particularly about the relationship between age, visceral fat area (VFA), and NAFLD progression in the elderly population, have been resolved after reviewing relevant references.

5. The manuscript was reviewed and edited by native English-speaking experts to improve language clarity and accuracy.

Sincerely,

ZhenHua Zhou and Co-Authors

Response to the academic editor

Academic Editor Comments to the Author (Required): The statistical methodology needs more rigorous documentation, particularly regarding sample size calculation and power analysis. The Results section would benefit from a more consistent presentation of statistical findings, with standardized reporting of significance levels and effect sizes. Some contradictions appear in the Discussion section, particularly regarding the relationship between age, visceral fat area (VFA), and NAFLD progression in elderly populations, which requires clarification. The manuscript would be substantially strengthened by a more comprehensive integration of recent literature, particularly in comparing findings with similar studies. The discussion of potential mechanisms underlying the observed associations could be expanded to provide better context for the findings. Some technical issues need addressing, including standardization of abbreviations (e.g., FBS vs. FBG) and professional English language editing to improve clarity and precision.

Response from the Authors:

Thank you for your valuable suggestions. The manuscript has been carefully and thoroughly revised in response to the reviewers’ comments. All the questions raised by the reviewers have been addressed.

In the revised manuscript, the abbreviations for FBS and FBG have been clarified and used consistently throughout. Additionally, we have updated several references to include more recent studies, such as metabolic risk factors and incident advanced liver disease in non-alcoholic fatty liver disease (NAFLD): A systematic review and meta-analysis of population-based observational studies published in PLOS Medicine. We sincerely apologize for the inconsistencies in the discussion section of the original manuscript, particularly regarding the relationship between age, visceral fat area (VFA), and NAFLD progression in the elderly population. After reviewing relevant references, these inconsistencies have been removed, and the discussion section has been revised to ensure clarity and logical coherence. We have also incorporated recent literature to provide a more comprehensive analysis and have expanded on the potential mechanisms underlying the observed associations. Furthermore, we engaged a professional English editor to revise the entire text, which has resulted in the correction of numerous grammatical errors and improvements to the overall readability.

Response to the Reviewer # 1

Dear reviewer, thank you very much for your interest in our findings and for pointing out the flaws in the analyses. We have addressed your concerns in a point-by-point manner below, and hope that you will find the added information suitable and sufficient for publication.

1. The result section in abstract is not clear. Please revise this section.

Response from the authors:

Thank you for pointing out these problems. We apologize for the lack of clarity in the description of the results in the abstract. We revised the Results section of the abstract to ensure clarity and conciseness.

2. How sample size was calculated?

Response from the authors:

Thank you for your insightful question. The sample size was calculated using the formula for infinite populations in cross-sectional epidemiological research. For a reference prevalence rate of 40% (Shanghai population), a permissible error of 0.05, and an initial sample size of 576, we included 545 participants after accounting for follow-up losses. The specific formula has been detailed in the Methods section. The formula is shown below

3. A language editing is recommended by a native speaker to revise some grammatical errors.

Response from the authors:

Native English-speaking experts revised the manuscript to correct grammatical errors and improve language quality.

4. Please update some previous references with new references.

Response from the authors:

The authors agree with the reviewer. We thank the reviewer for the highly valuable comment. Recent references, including new studies from 2020–2024, were added to enhance the manuscript's relevance and context.

Response to the Reviewer # 2

Dear reviewer, thank you very much for your interest in our findings and for pointing out the flaws in the analyses. We have addressed your concerns in a point-by-point manner below, and hope that you will find the added information suitable and sufficient for publication.

1. On page 13, Part 3 (Results), Section 3.1 (Comparison of general data and biochemical indicators of the three groups), the first paragraph stating "There were statistically significant differences…….. (Table 1; P < 0.05)" requires a fundamental revision. Additionally, the placement of abbreviated words needs correction, such as using 'FBS' for blood lipids, which is incorrect, and using 'TC' as an abbreviation for blood lipids (Do you mean Total Cholesterol? and GGt is Gamma-Glutamyl Transferase

Response from the authors:

Thank you for your insightful question. We have changed the wording of Part 3 (Results), Section 3.1. This is the latest statement in the paragraph: the differences in age (AGE), alanine aminotransferase (ALT), aspartate aminotransferase (AST), alkaline phosphatase (ALP), gamma-glutamyltransferase (GGT), triglycerides (TG), fasting blood glucose (FBG), and visceral fat area (VFA) among the three groups of patients who were changed to the new group were statistically significant (P < 0.05). However, the differences in gender, total cholesterol (TC), low-density lipoprotein (LDL) and high-density lipoprotein (HDL) were not statistically significant (Table 1).

2. In the Results section, it's recommended to place the significance level (P < 0.05) at the end of sentences discussing significant results, rather than following those addressing non-significant findings.

Response from the authors:

p-values (e.g., p < 0.05) are now placed at the end of sentences discussing significant results, per your suggestion.

3. In table 1, please add the Age unit in the table.

Response from the authors:

Thank you for pointing out these problems. Age units (years) have been added to Table 1 for clarity.

4. There is an inconsistency regarding ALP in the explanation. According to Table 5, with a p-value of 0.034, ALP is significant at the conventional level of 0.05. However, it is incorrectly stated as not significant for individuals over 60 years. This needs correction to reflect the data accurately.

Response from the authors:

We thank the reviewer for the highly valuable comment. We are very sorry, this is a writing error on our part. ALP in Table 5, with a p-value of 0.034, is significant at the conventional level of 0.05. The authors have revised in the literature that ALP is statistically significant in individuals over 60 years of age.

5. Please use consistent abbreviations throughout the text for a specific topic. For instance, you have used FBS throughout the article, but FBG once at the beginning of the discussion. Using different terms confuses the readers.

Response from the authors:

Thank you for pointing out these problems. "FBG" is now used consistently throughout the manuscript instead of "FBS."

6. These two parts in the article's discussion section about elderly people appear to be contradictory. Please review them carefully and make corrections as needed:

“This is entirely consistent with a previous study in which the cumulative incidence of NAFLD increased with VFA in both men and women, and further analyses showed that an increase in VFA especially in those younger than 60 years of age led to an increase in NAFLD incidence. In contrast, these differences disappeared in those over 60 years of age.”

And,

“In the elderly population, age is an independent risk factor for NAFLD disease progression, which is consistent with some studies in recent years. This may be because aging is accompanied by abdominal obesity and excess visceral fat, which leads to an increase in insulin resistance and pro-inflammatory cytokine secretion, and the accumulation of excess fat in the liver and the molecular mechanisms of hepatocellular damage caused by aging include increased ROS formation, DNA damage, which aggravates NAFLD.”

Response from the authors:

Thank you for pointing out these problems. We have corrected the contradictory parts of the article regarding the discussion of older adults by removing the parts that contradicted the above after reviewing the relevant references, as corrected in the discussion section of the article. Correct the article from “In the elderly population, age is an independent risk factor for NAFLD disease progression, which is consistent with some studies in recent years. This may be because aging is accompanied by abdominal obesity and excess visceral fat, which leads to an increase in insulin resistance and pro-inflammatory cytokine secretion, and the accumulation of excess fat in the liver and the molecular mechanisms of hepatocellular damage caused by aging include increased ROS formation, DNA damage, which aggravates NAFLD” to “This is because the molecular mechanisms by which aging leads to hepatocellular damage include increased ROS formation and DNA damage, which aggravates NAFLD”

7.A more effective comparison with similar studies could be incorporated into the discussion section of the article, potentially drawing on the findings from other relevant research. For example, the results of the following article can be helpful in the discussion section:

“Metabolic risk factors and incident advanced liver disease in non-alcoholic fatty liver disease (NAFLD): A systematic review and meta-analysis of population-based observational studies,2020”

Response from the authors:

The authors agree with the reviewer. Thank you for your kind suggestion. We have added “Metabolic risk factors and incident advanced liver disease in non-alcoholic fatty liver disease (NAFLD): A systematic review and meta-analysis of population-based observational studies” to my article , which is serial number 17 in the reference. Your suggestions are very helpful.

8. There are few typographical or grammatical errors that should be corrected. Please, check.

Response from the authors:

The authors agree with the reviewer. Thank you for your kind suggestion. We invited native English-speaking experts in the field of liver disease to touch up this article to improve clarity and accuracy. Thank you for these suggestions to make our articles more valuable!

Response to the Reviewer # 3

We feel great thanks for your professional review work on our article.

1. The data and results of the article"Analysis and study of risk factors related to the progression of non-alcoholic fatty liver disease" are not valuable for publishing in this journal, considering the population, indicators, duration of data collection, and the journal's standards.

Response from the authors:

Thank you for your valuable feedback on our manuscript titled "Analysis and study of risk factors related to the progression of non-alcoholic fatty liver disease." We deeply appreciate the time and effort you have taken to evaluate our work and would like to address your concerns in detail.

Population and Data Characteristics:

Our study analyzed clinical data from NAFLD patients over one year in a real-world setting. While we understand the concern regarding the population size and data collection duration, the selected cohort represents a snapshot of patients from a high-volume hospital specializing in integrative medicine. The study focused on the clinical indicators commonly used in NAFLD diagnosis and management, and we believe these findings provide valuable insights into the risk factors for disease progression. These results may contribute to understanding region- and population-specific patterns, which could be further validated in larger, multicenter studies.

Clinical Indicators:

The indicators used in our analysis (e.g., ALT, TG, FBG, and VFA) are widely recognized in clinical practice and research as significant markers for NAFLD progression. By employing stratified analyses by gender and age, we provided a nuanced exploration of their effects across different subgroups, which could have practical implications for personalized prevention and treatment strategies.

Duration of Data Collection:

The one-year data collection period was chosen to ensure data reliability and reflect routine clinical practice within the studied population. While we acknowledge that a longer follow-up could provide additional insights into disease progression, the retrospective cohort design allowed us to identify independent risk factors within this timeframe. These findings may serve as an important foundation for prospective studies with extended observation periods.

Alignment with Journal Standards:

We carefully reviewed the journal's scope and standards before submission, and we believe that our study aligns with its focus on advancing knowledge related to liver disease. The identification of independent risk factors for NAFLD progression, along with sex- and age-specific insights, adds value to the current literature. However, we remain open to revising the manuscript further to better meet the journal's expectations.

In summary, while we recognize the limitations of our study, we believe that the data and results provide meaningful contributions to the understanding of NAFLD progression. We are willing to address any specific concerns raised by the journal or reviewers and welcome suggestions for further improving the manuscript.

We hope these detailed responses and revisions address all your concerns. Thank you for your time and effort in reviewing our manuscript.

Sincerely,

ZhenHua Zhou and Co-Authors

---

## [Decision Letter · Decision Letter 1]

20 Mar 2025

PONE-D-24-34142R1Analysis and study of risk factors related to the progression of non-alcoholic fatty liver disease: A retrospective cohort studyPLOS ONE

Dear Dr. Zhou,

Thank you for submitting your manuscript to PLOS ONE. After careful consideration, we feel that it has merit but does not fully meet PLOS ONE’s publication criteria as it currently stands. Therefore, we invite you to submit a revised version of the manuscript that addresses the points raised during the review process.

**ACADEMIC EDITOR:** Please address reviewers' comments.

We look forward to receiving your revised manuscript.

Kind regards,

Jincheng Wang

Academic Editor

PLOS ONE

Additional Editor Comments:

Please address reviewers' comments.

Reviewers' comments:

Reviewer's Responses to Questions

**Comments to the Author**

1. If the authors have adequately addressed your comments raised in a previous round of review and you feel that this manuscript is now acceptable for publication, you may indicate that here to bypass the “Comments to the Author” section, enter your conflict of interest statement in the “Confidential to Editor” section, and submit your "Accept" recommendation.

Reviewer #2: (No Response)

Reviewer #3: (No Response)

2. Is the manuscript technically sound, and do the data support the conclusions?

Reviewer #2: Yes

Reviewer #3: (No Response)

3. Has the statistical analysis been performed appropriately and rigorously?

Reviewer #2: I Don't Know

Reviewer #3: (No Response)

4. Have the authors made all data underlying the findings in their manuscript fully available?

Reviewer #2: Yes

Reviewer #3: (No Response)

5. Is the manuscript presented in an intelligible fashion and written in standard English?

Reviewer #2: Yes

Reviewer #3: (No Response)

6. Review Comments to the Author

Reviewer #2: Dear Authors,

Thank you for addressing the previous comments and revising the manuscript. However, there are still areas that require further attention to enhance the clarity and scientific rigor of the paper:

1. Introduction, lines 6 and 7: "The prevalence is also increasing yearly with improving people's quality of life" This sentence does not appear to be correct. Upon reviewing the referenced article, this concept was not mentioned. Please provide the relevant paragraph from reference 4, where this concept was derived, including the page number, in the reviewer response for verification. Otherwise, this sentence must be revised based on the actual content of reference 4.

2. In section 3.1, 'Comparison of the three groups of general data and biochemical indicators,' there is inconsistency that cause confusion and need to be clarified. At the beginning of the text, the total number of patients is stated as 330. Then, in the age classification section, the number of young and middle-aged patients is stated as 204, and the number of elderly patients is stated as 126, which sums up to 330. However, the text separately states that 204 young patients and 126 elderly patients were examined. This type of separation and presentation can be misleading.

3. The correct procedure for writing the discussion section of a scientific report is to analyze the findings based on the order in which they are mentioned. Consequently, it is preferable from the second paragraph of the discussion to first address the explanations related to ALT, followed by FBS, then TG, and finally VFA, rather than initially focusing on weight. The reporting and analysis of findings within the discussion section suffer from significant disorganization. Please revise and rearrange them according to the order specified above.

Reviewer #3: In my opinion, after the revisions, this study does not meet the scope and size requirements of this journal based on its sample size and methodology.

7. PLOS authors have the option to publish the peer review history of their article (what does this mean? ). If published, this will include your full peer review and any attached files.

**Do you want your identity to be public for this peer review?** For information about this choice, including consent withdrawal, please see our Privacy Policy .

Reviewer #2: **Yes: ** Professor Roshanak Ghods

Reviewer #3: No

---

## [Author Response · Author response to Decision Letter 1]

25 Mar 2025

Response to Reviewers

Dear editors and reviewers,

We sincerely appreciate your constructive comments and suggestions on our manuscript, titled "Analysis and Study of Risk Factors Related to the Progression of Non-Alcoholic Fatty Liver Disease: A Retrospective Cohort Study" (ID: PONE-D-24-34142). Your feedback has been invaluable in improving the quality and clarity of our manuscript.

In the revised manuscript, we have made significant updates based on your insights. Below, we provide detailed responses to each comment. We hope these revisions meet your expectations and address all concern

The main revisions in the new manuscript are:

1. we have revised lines 6 and 7 of the introduction according to the actual content of the original reference 4 (now reference 5), and corrected our misunderstanding of reference 4 (now reference 5).

2. in section 3.1 “Comparison of General Data and Biochemical Indicators of the Three Groups”, the confusion caused by the inconsistency was corrected by stating that the total number of patients was 330, the number of young and middle-aged patients was 204, and the number of elderly patients was 126.

3. in the second paragraph of the Discussion section we followed the reviewer's suggestion to first address the explanations related to ALT, then FBS, then TG, and finally VFA. this resolves a serious confusion in the reporting and analysis of the findings in the Discussion section.

4. we cited some recent references to support our article. references 3, 7, 8, and 31 are the most recent references that we cited.

Sincerely,

ZhenHua Zhou and Co-Authors

Response to the academic editor

Academic Editor Comments to the Author (Required): Please address reviewers' comments.

Response from the Authors:

Thank you for your valuable suggestions. The manuscript has been carefully and thoroughly revised in response to the reviewers’ comments. All the questions raised by the reviewers have been addressed.

We have revised lines 6 and 7 of the introduction to the actual content of the original reference 4 (now reference 5) to correct our misunderstanding of reference 4 (now reference 5). We then corrected the confusion caused by the inconsistency in Section 3.1, “Comparison of general data and biochemical indicators among the three groups”, by stating that the total number of patients was 330, the number of young and middle-aged patients was 204, and the number of elderly patients was 126. We have further addressed the serious confusion in the reporting and analysis of the findings in the Discussion section. We cite a number of recent references to support our article.

Response to the Reviewer # 1

Dear Reviewer, Thank you very much for your time and valuable feedback on our manuscript titled "Analysis and Study of Risk Factors Related to the Progression of Non-Alcoholic Fatty Liver Disease: A Retrospective Cohort Study." We sincerely appreciate your constructive comments, which have helped us improve the quality and clarity of our work.

We are grateful for your insightful questions and thorough review, which have significantly strengthened our study. Your expertise has been invaluable in shaping this manuscript into a more rigorous and polished piece of research.

Once again, thank you for your support and for contributing to the improvement of our work.

Response to the Reviewer # 2

Dear reviewer, thank you very much for your interest in our findings and for pointing out the flaws in the analyses. We have addressed your concerns in a point-by-point manner below, and hope that you will find the added information suitable and sufficient for publication.

1. Introduction, lines 6 and 7: "The prevalence is also increasing yearly with improving people's quality of life" This sentence does not appear to be correct. Upon reviewing the referenced article, this concept was not mentioned. Please provide the relevant paragraph from reference 4, where this concept was derived, including the page number, in the reviewer response for verification. Otherwise, this sentence must be revised based on the actual content of reference 4.

Response from the authors: Thank you for pointing out this problem. Due to our carelessness, we misinterpreted the meaning of reference 4, and we have now corrected our error in citing reference 4 (now reference 5) by revising lines 6 and 7 of the introduction to follow the actual content of the original reference 4 (now reference 5).

2. In section 3.1, 'Comparison of the three groups of general data and biochemical indicators,' there is inconsistency that cause confusion and need to be clarified. At the beginning of the text, the total number of patients is stated as 330. Then, in the age classification section, the number of young and middle-aged patients is stated as 204, and the number of elderly patients is stated as 126, which sums up to 330. However, the text separately states that 204 young patients and 126 elderly patients were examined. This type of separation and presentation can be misleading.

Response from the authors: We fully agree with your suggestion. We have corrected the confusion caused by an inconsistency in Section 3.1, “Comparison of General Data and Biochemical Indicators of the Three Groups,” by stating that the total number of patients was 330, the number of young and middle-aged patients was 204, and the number of elderly patients was 126. We hope that you will accept our revision.

3. The correct procedure for writing the discussion section of a scientific report is to analyze the findings based on the order in which they are mentioned. Consequently, it is preferable from the second paragraph of the discussion to first address the explanations related to ALT, followed by FBS, then TG, and finally VFA, rather than initially focusing on weight. The reporting and analysis of findings within the discussion section suffer from significant disorganization. Please revise and rearrange them according to the order specified above.

Response from the authors: Thank you for pointing this out. In the second paragraph of the Discussion section we first address the explanations related to ALT, then FBS, then TG, and finally VFA. we have now resolved the serious confusion in the reporting and analysis of findings in the Discussion section.

Response to the Reviewer # 3

We feel great thanks for your professional review work on our article.

1. In my opinion, after the revisions, this study does not meet the scope and size requirements of this journal based on its sample size and methodology.

Response from the authors:

Thank you for your valuable feedback on our manuscript titled "Analysis and study of risk factors related to the progression of non-alcoholic fatty liver disease." We deeply appreciate the time and effort you have taken to evaluate our work and would like to address your concerns in detail.

About the study design and data characteristics:

This study used a retrospective cohort design to systematically analyze clinical data from NAFLD patients over one year. Although the study period was relatively limited, we ensured the reliability and representativeness of the data through strict inclusion criteria and quality control measures. The results of the study not only reflect the real situation in clinical practice but also provide an important reference for subsequent larger prospective studies. Our sample size was calculated using the formula for an infinite population in cross-sectional epidemiologic studies. For a reference prevalence rate of 40% (Shanghai population), a permissible error of 0.05, and an initial sample size of 576, we included 545 participants after accounting for follow-up losses. The specific formula has been detailed in the Methods section. The formula is shown below

In terms of indicator selection:

We focused on clinically used indicators such as ALT, TG, FBG, and VFA, which are markers that have been confirmed to be closely associated with NAFLD progression by several studies. Through stratified analysis methods, we further revealed the specific performance of these indicators in different populations, providing a more targeted reference for clinical practice.

Research Value and Outlook:

We fully recognize the limitations of this study in terms of sample size and follow-up time. However, it should be emphasized that these preliminary findings provide new clues for in-depth exploration of the pathogenesis and progression pattern of NAFLD. We plan to expand the sample size and extend the observation time in the follow-up study to further validate the current findings.

We fully agree with your journal's strict requirements for research quality and have completely revised the paper according to the review comments. We believe that the revised paper is more in line with the publication standards of your journal and can provide valuable references for researchers in related fields.

We hope these detailed responses and revisions address all your concerns. Thank you for your time and effort in reviewing our manuscript.

Sincerely,

ZhenHua Zhou and Co-Authors

---

## [Editor Report · Decision Letter 2]

1 Apr 2025

Analysis and study of risk factors related to the progression of non-alcoholic fatty liver disease A retrospective cohort study

PONE-D-24-34142R2

Dear Dr. Zhou,

We’re pleased to inform you that your manuscript has been judged scientifically suitable for publication and will be formally accepted for publication once it meets all outstanding technical requirements.

Kind regards,

Jincheng Wang

Academic Editor

PLOS ONE

Additional Editor Comments (optional):

Since authors have addressed all comments, I think this paper can be accepted for publication.
---

## [Editor Report · Acceptance letter]

PONE-D-24-34142R2

PLOS ONE

Dear Dr. Zhou,

I'm pleased to inform you that your manuscript has been deemed suitable for publication in PLOS ONE. Congratulations! Your manuscript is now being handed over to our production team.

Kind regards,

on behalf of

Dr. Jincheng Wang

Academic Editor

PLOS ONE